# Isolation and Characterization of Phages Active against *Paenibacillus larva**e* Causing American Foulbrood in Honeybees in Poland

**DOI:** 10.3390/v13071217

**Published:** 2021-06-23

**Authors:** Ewa Jończyk-Matysiak, Barbara Owczarek, Ewa Popiela, Kinga Świtała-Jeleń, Paweł Migdał, Martyna Cieślik, Norbert Łodej, Dominika Kula, Joanna Neuberg, Katarzyna Hodyra-Stefaniak, Marta Kaszowska, Filip Orwat, Natalia Bagińska, Anna Mucha, Agnieszka Belter, Mirosława Skupińska, Barbara Bubak, Wojciech Fortuna, Sławomir Letkiewicz, Paweł Chorbiński, Beata Weber-Dąbrowska, Adam Roman, Andrzej Górski

**Affiliations:** 1Bacteriophage Laboratory, Ludwik Hirszfeld Institute of Immunology and Experimental Therapy, Polish Academy of Sciences, Rudolf Weigl Street 12, 53-114 Wroclaw, Poland; barbara.owczarek@hirszfeld.pl (B.O.); martyna.cieslik@hirszfeld.pl (M.C.); norbert.lodej@hirszfeld.pl (N.Ł.); dominika8808@interia.pl (D.K.); joannaneuberg@gmail.com (J.N.); filip.orwat@hirszfeld.pl (F.O.); natalia.baginska@hirszfeld.pl (N.B.); barbara.bubak@hirszfeld.pl (B.B.); beata.weber-dabrowska@hirszfeld.pl (B.W.-D.); andrzej.gorski@hirszfeld.pl (A.G.); 2Department of Environment Hygiene and Animal Welfare, Wrocław University of Environmental and Life Sciences, Chełmońskiego Street 38C, 51-630 Wroclaw, Poland; ewa.popiela@upwr.edu.pl (E.P.); pawel.migdal@upwr.edu.pl (P.M.); adam.roman@upwr.edu.pl (A.R.); 3Pure Biologics, Duńska Street 11, 54-427 Wroclaw, Poland; kinga@purebiologics.com (K.Ś.-J.); k.hodyra@purebiologics.com (K.H.-S.); 4Laboratory of Microbial Immunochemistry and Vaccines, Ludwik Hirszfeld Institute of Immunology and Experimental Therapy, Polish Academy of Sciences, 54-427 Wrocław, Poland; marta.kaszowska@hirszfeld.pl; 5Department of Genetics, Wrocław University of Environmental and Life Sciences, Kożuchowska 7, 51-631 Wroclaw, Poland; anna.mucha@upwr.edu.pl; 6BioScientia, Ogrodowa Street 2/8, 61-820 Poznań, Poland; abelter@bioscientia.pl (A.B.); mirka@bioscientia.pl (M.S.); 7Institute of Bioorganic Chemistry, Polish Academy of Sciences, Noskowskiego 12/14, 61-704 Poznań, Poland; 8Department of Neurosurgery, Wrocław Medical University, Borowska 213, 54-427 Wrocław, Poland; wfortuna@onet.pl; 9Phage Therapy Unit, Ludwik Hirszfeld Institute of Immunology and Experimental Therapy, Polish Academy of Sciences, Rudolf Weigl Street 12, 53-114 Wroclaw, Poland; letkiewicz1@o2.pl; 10Department of Health Sciences, Jan Długosz University in Częstochowa, 12-200 Częstochowa, Poland; 11Department of Epizootiology and Clinic of Birds and Exotic Animals, Wrocław University of Environmental and Life Sciences, pl. Grunwaldzki 45, 50-366 Wroclaw, Poland; pawel.chorbinski@upwr.edu.pl; 12Infant Jesus Hospital, The Medical University of Warsaw, 02-006 Warsaw, Poland

**Keywords:** American foulbrood, honeybee, bacteriophages, phage isolation, *Paenibacillus larvae*

## Abstract

The aim of this study was the isolation and characterization, including the phage effect on honeybees in laboratory conditions, of phages active against *Paenibacillus larvae*, the causative agent of American Foulbrood—a highly infective and easily spreading disease occurring in honeybee larva, and subsequently the development of a preparation to prevent and treat this dangerous disease. From the tested material (over 2500 samples) 35 *Paenibacillus* spp. strains were obtained and used to search for phages. Five phages specific to *Paenibacillus* were isolated and characterized (ultrastructure, morphology, biological properties, storage stability, and genome sequence). The characteristics were performed to obtain knowledge of their lytic potential and compose the final phage cocktail with high antibacterial potential and intended use of future field application. Preliminary safety studies have also been carried out on healthy bees, which suggest that the phage preparation administered is harmless.

## 1. Introduction

Pollination is the most important process necessary for obtaining seeds for many plants. This means that lowering its efficiency would result in the depletion of the plant flora across the whole planet. Globally, 87% of major food crops depend on animal pollination [1]. Insect pollinators, in particular, play a dominant role in this process [2,3], in which bees are one of the most important groups, which is necessary for the development of 75% of all crops that are used directly to produce food for humans worldwide [4]. In 2005, the global annual economic value of insect pollination was estimated to be 153 billion euros [5]. Around 84% of European crops benefit, at least in part, from insect pollination. In 2005, around 10% of the total economic value of European agricultural output for human food, amounting to €22 million, was dependent upon insect pollination [6].

A progressive decrease in the number of *Apis mellifera* (honeybee) colonies has been observed worldwide in the last few decades [7,8] which has caused a decrease in agricultural production all over the world. American foulbrood (AFB) is suggested as one of the main possible causes, which is a dangerous bacterial disease caused by Gram-positive bacterium—*Paenibacillus larvae*. It is an easily spread brood infection with worldwide distribution [9], dangerous for honeybees, particularly European [10,11]. Due to its high pathogenic potential and difficult elimination process in the apiary, this disease may cause entire colony death [12]. The affected hives undergo a progressive depopulation and, as a result of a general weakening effect, colonies become more susceptible to robbery by stronger and uninfected colonies from the same apiary or from neighboring ones. In this way, disease spread is favored [13]. The high infectivity of the disease is associated with the fact that the causative pathogen is spore-forming bacterium [14], and the spores may persist even for several decades in the hive environment due to extreme resistance to external conditions as well as chemical agents [10,15]. This is the reason why applying conventional therapies is not effective in the fight against AFB, because the vegetative forms of bacteria are destroyed, but the applied methods are ineffective against spores (transferred within and/or between colonies) [16,17]. The use of antibiotics may induce antibiotic resistance in the case of bacterial cells [18]; moreover, antibiotic residues may be detected in honey [19,20] and nowadays application of antibiotics in the fight against AFB has been banned in most of the European Union [21]. In many countries, disease control even includes the burning of infected colonies, which generates huge economic losses.

The development of an effective and safe method for the prevention and control of AFB is highly desirable. Bacteriophages may be a promising solution in this area [22,23]. These bacterial viruses have proven to be common in the hive environment [22,24,25,26,27,28,29,30,31,32,33] and participate in bee gut microbiota [34], modulating its structure as well as influencing honeybee health [35]. The unique feature of phages is their ability to act with high specificity exhibiting lytic activity only against their bacterial host as well as amplifying at the site of infection [36]. Additionally, there are reports of the possible use of phages in the food industry as factors contributing to increasing the safety of food production, both of animal and plant origin, and preventing the spread of pathogenic bacteria [37,38].

Published data regarding this topic have been scant and have indicated that the specific isolated phages showed the ability to lyse only the vegetative forms of *P. larvae* strains, and in vivo effects have suggested that phages could be particularly useful in AFB prevention rather than treatment [30,39], whereas current data suggest that *P. larvae* phages may also bind to vegetative and spore forms of host bacteria, and binding spores may be reversible, which has been suggested as having great potential in the treatment of infection caused by sporulating bacteria [40]. Further research on the practical application of *P. larvae* phages regarding, e.g., different phage titers, different phage formulation compositions, forms of preparation, application mode, and addition of carriers both in vitro and in vivo during different stages of bee development are urgently needed [23].

From the procured and studied material we isolated and characterized five phages specific to *Paenibacillus* whose safety after application in healthy bees is described in this paper.

## 2. Material and Methods

### 2.1. Samples Used for Isolation Bacteria and Phage Procurement

Over 2500 biological and environmental material samples, such as soil (*n* = 189), honey (*n* = 516), bees (*n* = 362), wax comb (*n* = 680), bee brood (*n* = 266), hive swabs (*n* = 325, including swabs from bottom board *n* = 28), bacterial cultures (*n* = 82), and bee pollen (*n* = 148), were used to isolate the bacterium that caused AFB and specific phages. The samples were collected from different areas of Poland and were delivered in both the 2017 and 2018 beekeeping season. They came from both healthy bee colonies and those suspected of infection with *P. larvae*. Additionally, isolated stains of *P. larvae* came from the Veterinary Hygiene Institution, Katowice. Additionally, samples (*n* = 251) from municipal and hospital sewage, farm animal surroundings (poultry and pigs), and natural and artificial water reservoirs collected in the Bacteriophage Laboratory were used. Standard strains, *Paenibacillus larvae* subsp. *larvae* (ATCC-9545), *Paenibacillus larvae* (NRRL B-3685ATCC-49843), *Kurthia* spp. (ATCC-39312), and *Paenibacillus alvei* (ATCC-6344), were used during the optimization of bacterial culture methods, as well as for research related to bacteriophage search and determination of their lytic spectrum.

### 2.2. Bacterial Culture

Biological and environmental samples after preparation (depending on the type of processed material) and weighing, were suspended in a MYPGP culture medium (for culture *Paenibacillus* spp. strains) and in Brain Hearth Infusion (BHI) or phosphate buffered saline (PBS) [41]. One culture was incubated in a heating bath (6 min, 70 °C) in order to destroy the spore forms and to allow germination of spores of persistent bacteria (mainly *P. larvae*). The second was incubated at 34 °C from 24 h to 5–7 days with shaking (150 RPM). Bacterial suspensions were poured onto plates with MYPGP and BHI solid media. A streak culture was carried out to obtain pure growth of the obtained cultures. Then they were suspended in MYPGP or BHI liquid medium with the addition of 20% glycerol (Difco) and frozen or placed in a semi-liquid medium under paraffin to protect against possible mutational changes.

### 2.3. Identification of P. larvae Strains

Rapid AFB (Vita) tests, which detect the reaction of specific antibodies previously associated with stained latex particles, allowing the presence of *P. larvae* to be detected in tested inoculum, were used [42]. Strains that were initially identified as *P. larvae* were tested using a MALDI-TOF/TOF mass spectrometer (Bruker Daltonics). This is currently the most sensitive and the most reliable method of identifying microorganisms, so species belonging to the *Paenibacillus larvae* were identified using this method [43], based on the analysis of proteins present in large quantities, such as ribosomal proteins. A material containing 10^5^ bacterial cells (single colony) is sufficient for identification. Bacterial colonies were applied to a steel plate, followed by the addition of 1 μL matrix (*α*-cyano-4-hydroxycinnamic acid (HCCA) dissolved in a mixture 500 μL acetonitrile (AcN), 450 μL water and 25 μL trifluoroacetic acid (TFA), and dried at room temperature. The mass spectra obtained for a given strain were compared each time with the reference spectra of the MALDI BioTyper 3.1 database (Bruker Daltonics, Billerica, MA, USA) in the molecular weight range of 2–20 kDa proteins. Mass spectra were obtained for each tested strain and compared with the spectra of the reference strains.

### 2.4. Phage Isolation

The tests were carried out using the plate method (phage typing, Ślopek et al., 1983) [44], consisting of preparing a suspension of an appropriate bacterial strain in a liquid MYPGP medium (incubation 3–4 h, 34 °C, 150 RPM) until the bacteria were obtained in the logarithmic phase growth (culture density ~1 in the McFarland scale). Next, the inoculum was poured and spread on a plate surface with a solid medium and left to dry (1 h, 34 °C). Drops of samples (~25µL) for phage searches were spotted on the surface of the solid MYPGP medium with bacterial lawn. After the plates were dried and incubated (at 34 °C, 18 h), the results were checked and calculated. The presence of a zone of inhibition of growth (lysis) or single “plaques” in the bacterial lawn indicated the probability of the presence of phages in the tested material. The results were evaluated according to the scheme developed by Ślopek et al. (1972) [45]. Samples causing bacterial lysis were additionally diluted from 10^−1^ to 10^−4^ and spotted on plates, like previously obtained plaques, and confirming that lysis was the result of phage activity. Phages in biological and environmental material were also searched for, using amplification in the enrichment culture [46]. The bacterial culture was incubated with the phage screened sample, then the cultures were filtered through a 0.22 µm bacteriological filter (Merck) and tested with the plate method. At the same time, all the *Paenibacillus* spp. strains were checked for the ability to release prophages [47]. An overnight culture of the bacterial strain (broth culture) was inoculated in a liquid MYPGP medium, 0.5 μg of mitomycin C per 1 mL was added and incubated (4 h, 37 °C), followed by centrifugation (1400× *g*, 20 min). The resulting supernatant was applied on plates with solid medium and incubated (18 h at 34 °C) and the appearance of phage plaques on the medium surface was checked.

Host *P. larvae* strain was suspended in MYPGP medium and incubated for 18 h at 34 °C. 100 µL of bacterial suspension and 200 µL of the filtered and diluted sample with confirmed phage presence was added to 2 mL of melted 0.7% agar, mixed together on vortex and poured over the surface of a MYPGP plate. Plates were left to solidify for 30 min and checked for plaque appearance after an overnight incubation at 34 °C. The single plaque was picked, added to 10 mL of MYPGP medium and left for 3 h to spread in the medium. Then 50 µL of host *P. larvae* suspension was added and incubated overnight at 34 °C. Next, the suspensions were filtered with the use of bacteriological filters (0.22 µm membrane pores) to separate amplified phages from bacterial cells. Obtained lysate was tested for phage homology by checking plaque morphology with the use of the Adams (1959) method [48]. If necessary, the process of plaque isolation was continued. If there was a need to amplify the phages on a larger scale, then *P. larvae* strain was cultured on MYPGP agar (34 °C, 5% CO_2_, 48 h). Next, bacteria were suspended in 10 mL of MYPGP liquid medium and incubated (18–24 h, 34 °C). Suspension was diluted in a fresh MYPGP medium to obtain an optical density of 0.3 at 600 nm wavelength (OD_600_). For bacteriophage enrichment, 100 µL of inoculum was added to 400 mL of MYPGP medium and incubated (18 h, 34 °C, 0 RPM) and then 50 µL of phage lysate was added and incubation was continued (24 h, 34 °C, 150 RPM). To obtain lysate, the culture was filtered (0.22 µm). After filtration, the lysate was checked for any contamination by incubation (48 h, both at 34 °C and 22 °C).

### 2.5. Phage Titer Determination and Plaque Morphology Assessment

To determine the tested phage lysate’s titer, Routine Test Dilution (RTD) and double layer agar method [48,49] were used. Routine test dilution (RTD): *P. larvae* host strain was suspended in a MYPGP medium and incubated (18 h, 34 °C). Next, the inoculum was spread onto the MYPGP agar plate and incubated (60 min, 34 °C). Phage lysate was diluted in liquid MYPGP medium from 10^−2^ to 10^−6^ and 25 µL of each dilution was spotted on MYPGP agar plates with bacterial lawn. The plates were dried and the appearance of areas of growth inhibition on the surface of bacterial lawn after overnight incubation (34 °C) was checked. The phage titer (expressed as plaque forming units per milliliter: PFU/mL) of the lysate was determined on plates which contain a countable number of plaques; Double layer agar method: This method was used to determine the phage lysate’s titer, as well as the plaque’s morphology, more precisely. Briefly, *P. larvae* host strain was suspended in a MYPGP medium and incubated (18 h, 34 °C). 100 µL of bacterial suspension and 200 µL of the phage lysate’s dilution was added to 2 mL of melted 0.7% agar, mixed and poured on the surface of a plate with MYPGP solid medium. Three replicates of each dilution were made. Plates were left to solidify for 30 min and incubated overnight (34 °C). Determination of the titer of phage lysate was made by using the following calculation: number of plaques × 5 × reciprocal of counted dilution = PFU/mL.

### 2.6. Phage Specificity (Host Range)

The sensitivity of bacteria associated with *P. larvae* strains (i.e., isolated from the intestines of healthy bees not infected with AFB) to tested bacteriophages was evaluated. Among these, the following lactic acid bacteria were distinguished: *Lactobacillus fructivorans*, *Lactobacillus curvatus*, *Lactobacillus kunkeei*, *Lactococcus lactis*, *Lactobacillus helveticus*, *Lactobacillus plantarum*, *Leuconostoc mesenteroides*, *Lactobacillus paracasei*, *Lactobacillus delusiaceus*, and *Lactobacillus delbruechei*, and other strains: *Hafnia alvei*, *Klebsiella oxytoca*, *Citrobacter koseri*, *Morganella morganii*, *Klebsiella aerogenes*, *Proteus vulgaris*. Liquid cultures were prepared in the MYPGP medium and incubated (34 °C, 150 RPM, 24 h). Next, bacterial inoculum was poured onto the plate with solid MYPGP medium to cover the entire agar surface, and the plates were dried for approximately 1.5 h, then the drop (~25 μL) of phage lysate was spotted to the dried plate and incubated (depending on the tested strain: 34 or 37 °C, 18–24 h). The results were then checked. Plates were prepared in triplicates.

### 2.7. Phage Lytic Spectra

The lytic spectrum studies of the isolated bacteriophages were tested on 35 bacterial strains confirmed as belonging to *Paenibacillus* spp. (standard strains, strains isolated from environmental samples such as wax comb, bee brood). For the first step, a liquid bacterial culture suspended in a MYPGP medium of each *P. larvae* strain was prepared and incubated (34 °C, 150 RPM, 24 h). Next the bacterial inoculum was poured on plates with a solid MYPGP medium to cover the entire agar surface, and then the excess culture was drained. The plates were dried and a drop of each tested lysate was spotted on the bacterial lawn and incubated (34 °C, 24 h). The appearance of bacterial growth inhibition (lysis) indicated the susceptibility of the bacterial strain to the tested phage. The results were checked on the basis of the scheme proposed by Ślopek et al. (1983) [44]. Plates were prepared in triplicate.

### 2.8. Determination of Endotoxin Level in the Tested Preparations and Phage Purification Procedure

The level of bacterial endotoxin contamination was determined using the Limulus Amebocytic Lysate Chromogenic (LAL) assay (Lonza, Switzerland). Principle of the test: the endotoxin of Gram-negative bacteria catalyzes the activation of the proenzyme in the Limulus Amebocytic Lysate (LAL). The activated enzyme catalyzes the cleavage of the colorless substrate Ac-Ile-Glu-Ala-Arg-pNA. The presence of the pNA molecule is measured photometrically at a wavelength of *λ* = 405–410 nm after the reaction is stopped by a stop reagent. The correlation between absorbance and endotoxin concentration is linear in the range 0.1–1.0 EU/mL. The endotoxin concentration in the sample is calculated using the absorbance value of solutions containing known amounts of standard endotoxin. A microplate method is used, in which the reaction is carried out in a heating block at 37 °C ± 1 °C and after the reaction is completed, the results are read on a spectrophotometer. Phage lysates were purified by size exclusion chromatography using a Sepharose CL-4B 26/100 column (GE Healhcare, Chicago, IL, USA). The matrix was 4% cross-linked agarose. A sample volume of 15 mL was loaded onto the column at a flow rate of 1.3 mL/min and eluted with a phosphate-buffered saline (PBS). Phage titer was determined with the double layer agar method. Samples were prepared in triplicate.

### 2.9. The Effects of Storage Conditions, Addition of Sucrose and Royal Jelly on Phage Activity

Stability of the purified preparations and crude lysates of phage 1/A, 2/A, 3/A and 5/A at three temperatures, 4 °C, room temperature (approx. 22 °C) and 35 °C (similar to that in the hive), was studied. The phage lysate was pre-dialyzed into a PBS buffer containing the remnants of the medium on which the phages were amplified (MYPGP), while the purified preparation contained phages suspended in PBS buffer and purified by size exclusion chromatography using a Sepharose CL-4B column. The experiment lasted 147 days. The titer of the preparations was checked using double layer agar method.

The stability of the P3 preparation (1/A + 3/A) was further tested at 22 °C and 4 °C. The experiment was carried out for 254 days. The titer of the tested preparations was determined with the use of the double layer agar method.

In subsequent experiments, the research on the previously selected 50% sucrose solution was extended. The stability of 1% and 10% phage preparations: P1 (1/A), P2 (3/A) and P3 (1/A + 3/A) suspended in 50% sucrose were stored at room temperature and at 4 °C, over 70 and 75 days. The effect of 5% royal jelly on phage activity was checked over 12 days at 4 °C.

### 2.10. Determination of Optimal Multiplicity of Infection—Optimization of Phage Amplification Process

Liquid bacterial culture of the host strains (*Paenibacillus thiaminolytics* No. 408 and *Paenibacillus larvae* 7030T No. 453), previously incubated in the MYPGP broth (34 °C, 18–24 h), were added to the 30 mL of this broth and incubated under the same conditions. In order to determine the optimal multiplicity of infection (MOI) for all tested phages, 1/A, 2/A, 3/A, 4/A, 5/A, which is defined as the ratio of infectious virions to cells in a culture, variable culture conditions (temperature, time and RPM) were used. The aim was to find the MOI at which the obtained phage titer would be the highest for the tested phage. For example, the lysate of 1/A phage with a titer of 10^8^ PFU/mL was added to the bacterial culture, and incubated (24 h, at 34 °C or 37 °C with 150 RPM or 0 RPM). After filtering the lysates through bacteriological filters (0.22 µm), their titer was determined by the double layer agar method. The efficiency of amplification of *P. larvae* phages was checked with a volumetric MOI of 1:1, 1:10, 1:100, 1:1000, 1000:1, 100:1, 10:1. The titers of the bacterial suspensions were determined using a spectrophotometer at *λ* = 600 nm. Phage titers after culture were determined by an RTD test in triplicate.

### 2.11. Phage Adsorption Rate In Vitro

The adsorption of phages to bacterial cells was examined according to the method described by Roncero et al. (1990) [50] and Gallet et al. (2009) [51]. Equal volumes of an overnight bacterial suspension (~10^6^ cells/mL) and phage (titers 10^5^ PFU/mL) were incubated at 34 °C and 150 RPM for 10, 20 and 40 min. After incubation, the collected samples were filtered (0.22 μm) and the number of plaques in the filtrate was determined using the RTD and the double layer agar method. The density of the bacterial culture was examined by spectrophotometric method. Then the percentage of adsorbed phages was calculated. Plates were prepared in triplicates.

### 2.12. Phage Lyophilization with Addition Sugar as Cryoprotectant

The tested phages, 1/A, 2/A, 3/A, 5/A, and cocktail of two phages, 1/A and 3/A, were used. The phages were purified by size exclusion chromatography using a Sepharose CL-4B column and suspended in miliQ water (method described previously). Phage 4/A was excluded from the research, because of a problem with obtaining a high titer in the purified preparation. Phage titer was checked with the use of the double layer agar method. The tested phages were divided into 5 parts for 10 mL. To each part appropriate sugar was added to 10% of the final volume (*w*/*v*). As a negative control, phages without addition of sugar were used. The formulations were mixed by conversion until the sugar was completely dissolved (30 min). Then phages were divided on 0.5 mL samples and stored at −80 °C. Frozen samples were opened and temporally sealed with pierced lids then subjected to lyophilization for 48 h in Christ Alpha LS-4 Freeze Dryer (parameters: trap-chamber temp: −60 °C; shelf temp −20 °C; vacuum 1.03 mbar; process length—48 h) (modified method described by Manohar et al., 2019 [52]). Dried samples were resealed with original caps and wrapped with parafilm. One phage portion with appropriate sugar was resolved in 0.5 mL miliQ water (the same volume) and the phage lysate’s titer was monitored 40, 75, 100 and 125 days after lyophilization.

### 2.13. Electron Microscopy Analysis 

Phage lysate a with titer of 10^7^ PFU/mL was purified and concentrated by ultracentrifugation at 25,000× *g* for 1 h at 7 °C. Supernatant was removed, 0.1 M ammonium acetate was added to residue and mixture centrifuged as previously. The process was repeated if necessary. A drop of phage suspension was deposited on a 400 mesh copper grid (Athene) and stained with 2% uranyl acetate. Dried preparations were examined in a Zeiss EM900 TEM at an acceleration voltage of 80 kV. Photos were taken on Kodak/Carestream Electron Microscope Film 4489. Based on the obtained images, the morphology, sizes and taxonomy of tested phages were determined [53,54].

### 2.14. Phage DNA Isolation and Sequencing

Before DNA isolation 20 mL of phage lysates with appropriate titers: 1/A—8 × 10^5^ PFU/mL; 2/A—3 × 10^8^ PFU/mL; 3/A—8 × 10^6^ PFU/mL; 4/A—4 × 10^5^ PFU/mL; 5/A—4 × 10^5^ PFU/mL were concentrated (6000× *g*) using a Vivaspin centrifugal concentrator with a 10kDa membrane (MERCK; Cat. no. UFC901096) to the final volume of 1–2 mL. Phage DNA was isolated from 1 mL of phage lysates using a commercial Phage DNA Isolation Kit (Norgen Biotek, Thorold, ON, Canada; Cat.no. 46850) and DNA from bacterial strains 408 and 453 were isolated from 1 mL of bacterial culture by NucleoSpin Microbial DNA Mini kit for DNA from microorganisms (MACHEREY-NAGEL, Düren, Germany; Cat.no.740235.50) according to the manufacturer`s protocols. Whole genomic DNA was sequenced using NGS (ang. Next Generation Sequencing) on the Illumina HiSeq by BaseClear Company (https://www.baseclear.com/, 15 January 2019).

### 2.15. Bioinformatics Analyses of Sequence Data

De novo analyses of genomic sequences were performed. Host sequences were extracted and deleted from phage sequence data. DNA readings were assembled into the complete genome of bacteriophages using four bioinformatic tools. The Artemis software reads files in GenBank and EMBL database extensions, and indicates potential open reading frames (ORFs) on one of the six reading frames, specifying their minimum length. Artemis allows us to edit ORFs, enter descriptions in accordance with the database standard and search for homologues using BLAST (Basic Local Alignment Search Tool) in the GenBank database. BLAST is a program which focuses on regions containing homologous sequences with available sequences in the database. Depending on the sequence, BLASTn can be used based on nucleotide sequence, or BLASTp based on amino acid sequence. Characteristic sequences (high-scoring segment pairs) are searched by algorithms and compared with the sequences available in the database; pfam—database contains sets of protein families defined by sequence comparisons and HMM (Hidden Markov Model) algorithms. Each of the database entries is based on a reference to the UniProt database, which contains complete information on proteins expressed by fully sequenced organisms; EasyFig is a program used to create linear maps and comparisons between genomes or gene sets. Based on the annotation, a linear genomic map was created for each of the phages, in which the most important genes were marked.

### 2.16. Sample Preparation for MALDI-TOF Analysis of Phage Protein

The phage samples (1/A, 2/A, 3/A and 5/A phages) were analyzed using polyacrylamide gel separation (SDS-PAGE) (Appendix A). Then all bands were cut out from the gel with a clean scalpel. The gel pieces were transferred into a microcentrifuge tube and to remove Commassie reagents from the gel, 100 mM ammonium bicarbonate/acetonitrile (1:1, *vol*/*vol*) was added and incubated for 30 min. 500 mL of neat acetonitrile was added to tubes and incubated at room temperature with occasional mixing, until gel pieces became white and shrank and then acetonitrile was removed. Trypsin buffer (13 ng/µL 1 trypsin in 10 mM ammonium bicarbonate containing 10% (*vol*/*vol*) acetonitrile) was added to cover the dry gel pieces. Gel pieces were left for another 90 min to saturate them with trypsin and then ammonium bicarbonate buffer (10 mM dithiothreitol (DTT) in 25 mM ammonium bicarbonate with 10% acetonitrile) was added to cover the gel pieces and keep them wet during enzymatic cleavage. Tubes with gel pieces were incubated overnight at 37 °C. A volume of water equal to two or three times of excised gel pieces was added to the tubes. The tubes were vortexed for 10 min, sonicated for 5 min, and centrifuged. The supernatants were transferred into a fresh tube and 45% water/50% acetonitrile/5% formic acid were added to the tube containing the gel pieces so that the pieces were fully immersed and vortexed for 10 min, sonicated for 5 min, and centrifuged briefly. The supernatants were transferred into a fresh tube. These steps were repeated twice. The sample volume was reduced to 10 µL using a vacuum centrifuge prior to submitting samples to analysis by LC-MS. The samples were analyzed using MALDI-TOF mass spectroscopy. After obtaining data from MALDI-TOF, the Swissprot database (SwissProt, Geneva, Switzerland, 2017 (553.655 sequences; 198.177,566 residues) was searched using ProteinPilot (AB Sciex, Framingham, MA, USA) and MASCOT Version: 2.3.0. (Matrix Science, Chicago, IL, USA). Moreover, data obtained by MALDI-TOF were compared to the genomic sequences from phages 1/A, 2/A, 3/A and 5/A.

### 2.17. Effect of Phage Cocktail on Healthy Adult Bee Survival

An in vivo model was used to rule out the possible negative effect of the selected phages on adult bees. The studied material consisted of honeybee workers (*A. mellifera* L.) of the Carniola race. Worker bees were collected from the border frames of the brood part of the 3 healthy colonies and transported to the laboratory. In the laboratory bees were distributed to experimental cages. Identical cages of wood and glass with dimensions of 50 × 150 × 150 mm were supplied with two 5 mL feed dispensers containing test or control solutions. Each group contained 9 cages (*n* = 9). The total number of bees in the control group (K), P1-1%, P1-10%, P2-1%, P2-10%, P3-1% and P3-10% group was 742, 889, 872, 732, 767, 767 and 783, respectively. The cages were put in an incubator where constant temperature and humidity (T = 34 ± 0.5 °C and H = 65 ± 5%) were maintained. The bees were fed ad libitum with 50% sugar syrup (1:1 *w*/*v*) during the first 24 h dedicated to acclimatization to the new environmental conditions. After this time cages were divided into 7 groups: control group (K)—fed only with 50% sugar syrup; group P1-1%—fed sugar syrup with the additive of 1% of phage 1/A, group P1-10%—fed sugar syrup with addition of 10% of phage 1/A, group P2-1%—fed sugar syrup with supplementation of 1% of phage 3/A, P2-10%—fed sugar syrup with supplementation of 10% of phage 3/A, P3-1% and P3-10% fed with sugar syrup with an addition of 1% and 10% of the phage cocktail 1/A + 3/A (P3), respectively. Every day the solutions in feed dispensers were replaced with new ones in order to minimize the risk of biological changes in the tested phage cocktails. The number of dead bees and feed intake were recorded daily for 7 days. Finally, mean feed intake (µL of syrup/bee/day) and mean bee mortality (%) were calculated.

By using Noldus Observer XT software the behavioral analyses were calculated. In this case, the total number of 147 bees from all groups were observed. Each bee was observed for 5 min. in the observing area. The observing area was constructed from a glass ring with a plexiglass cover and the camera mounted above it to record films. The following behavioral factors were taken into account: walking, grooming, flying, trophallaxis, immobility, and fanning. An assessment was made based on frequency and time of duration of a given behavior. The mean time of each behavior (s) and the mean number of each behavior in the group were calculated.

### 2.18. Effect of Phage Cocktail on Honey Bee Larvae Survival

Experiments were conducted on larvae obtained from colonies headed by naturally mated queens kept in the apiary according to standard industry practices. First instar larvae were collected from worker brood cells. Larval age was controlled by confining the queen to empty combs and controlled egg stadium change on the first instar larvae stadium. Larvae were transferred and reared at high humidity (90.0 ± 5.0%) and controlled temperature (34.0 ± 0.5 °C) in plastic queen cups containing an excess of a liquid diet. Feed for larvae was prepared, consisting of distilled water, royal jelly powder, glucose, fructose and yeast extract. To accommodate protein and sugar requirements of developing larvae, the composition and amounts of larval feed was changed based on the larval stage [55].

All ingredients were mixed to ensure complete dispersion. The fresh diets were prepared and stored at 4 °C for less than two days during the experiment. Prior to the feeding, diets were pre-warmed at 34 °C and given to the larvae according to their age using a pipette. In order to avoid possibility of contamination which may be result of superinfection by other bacteria and to reduce the effect of the possible effect of royal jelly pH on phage stability in queen cups, feed was refreshed every 24 h. If there was any feed left in the cup, it was carefully removed with a pipette before giving a fresh portion of feed.

Every 24 h larvae were monitored for survival under a dissecting microscope. Larvae were classified as dead by the absence of respiration or movement, color change, edema and loss of body elasticity [56,57]. In the experiment larvae were divided into three groups: negative control (K1)—larvae were fed without any additives (only basic larval food), K2—control group fed larval food with addition of PBS and LP3—group fed larval food reaching phage cocktail suspended in PBS a final concentration of 1 × 10^5^ PFU/mL. Each group contained a total of 80 larvae per treatment.

### 2.19. Statistical Analysis

The statistical analysis was performed using R Project 3.5.3 [58]. The basic descriptive statistics of considered traits were determined with the pastecs package [59]. The compliance of the distribution of analyzed traits with the normal distribution was verified using Shapiro-Wilk test. The statistical significance of differences in mean bee mortality, feed intake and behavioral factors between particular groups was verified with Kruskal-Wallis non-parametric analysis of variance in the agricolae package [60]. The significance of differences in larvae mortality for particular days of the experiment, due to the consistency of the data distribution with a normal distribution, was verified using the Student’s *t*-test for dependent samples with the Bonferroni correction. Statistical significance was determined when *p* < 0.05.

## 3. Results

From the tested material 35 bacterial strains (presented in Table 1) were identified as those belonging to genus *Paenibacillus* such as: *P. larvae* (*n* = 29), *P. pabuli* (*n* = 2), *P. chinjensis* (*n* = 1), *P. alvei* (*n* = 1), and *P. thiaminolitycus* (*n* = 1). These strains were used for further studies. Over 2500 samples were searched for phage using different methods and five phages specific to *Paenibacillus* spp. were obtained and characterized. The main sources of phage isolation were wax combs and bees (Table 2). The morphology of the observed plaques formed by each of the tested phages was similar. The diameter was approx. 0.5 mm, and they had a transparent, slightly irregular edge, especially easily visible on the bacterial lawn on the solid culture medium after determining the phage titer using the double layer agar method. The halo effect was initially invisible, appearing after about 3 days. Each of the tested phages has individual requirements regarding amplification conditions. *P. larvae* host strains for mixed cultures were selected on the basis of their sensitivity to a given phage and growth rate. Growth curves were plotted by measuring the optical density (OD) at a wavelength of *λ* = 600 nm while culturing the bacterial strain in a liquid medium (34 °C and 0 RPM). The results of the measurements made it possible to distinguish two groups of strains for which the logarithmic growth phase fell on, respectively, 10–18 h of culture and 22–35 h of culture. Five isolated bacteriophages were amplified only in an enriched culture.

The sensitivity of *P. larvae* strains to isolated bacteriophages expressed as a percentage was: 31–46% (sensitivity of 16/35 strains), for 1/A: 46%, 2/A: 46%, 3/A: 46%, 4/A: 31% (11/35 strains susceptible), 5/A: 46%. Phages 1/A, 2/A, 3/A and 5/A had a similar lytic spectrum and pattern of lysed strains, while 4/A was less susceptible. Our results show that the bacteriophages are specific only to strains belonging to *Paenibacillus* spp. Other species of bacteria were not sensitive to the tested phages. Additionally, the susceptibility of probiotic strains present in the bees’ intestines to the described phages was not observed.

Studies using TEM have shown that all *P. larvae* phages belong to the *Siphoviridae* family and an elongated shape, consisting of a head and a long non-contractile tail. Phages 1/A–4/A belong to the B2 morphotype—they have a short tail, while 5/A represents B3 morphotype which is characterized with longer tail. Figure 1A–E show the electrono-grams of phages 1/A–5/A. The morphological characteristics of the phages studied, established on the basis of observations in TEM, are presented in Table 3.

After phage characterization, the most effective were identified to compose a cocktail intended to use as prevention and or treatment of AFB in honeybee larvae. We used different phage preparations, single phage as well as cocktails, to check and compare their activity and safety of use. Preparation used in the described experiments: P1 contained 1/A phage suspended in PBS with a titer of 6.2 × 10^6^ PFU/mL, P2 contained 3/A phage suspended in PBS with a titer of 8.8 × 10^6^ PFU/mL, P3 contained cocktail of 1/A and 3/A phage in a 1:1 volume ratio, final titer of each phage was 1 × 10^6^ PFU/mL. The titer in the final volume of preparations was, for the 10% preparation ~10^5^ PFU/mL and, for the 1%, ~10^3^ PFU/mL.

In the purified *Paenibacillus*-specific phage preparations, the endotoxin levels depended on the tested phage and were 0.24–2.5 EU/mL (Table 4). The level of bacterial endotoxins of individual phages (1/A and 3/A) included in the bacteriophage cocktail in selected samples of the preparation used in bees was also determined; for 1/A phage 0.570–0.873 and for 3/A 0.889–0.969 EU/mL.

Phages in lysate and purified form (except 4/A) stored at 4 °C maintained their lytic activity till the end of the experiment (Figure 2). When phage cocktail kept at room temperature they showed a quick decrease in their titer. The differences in mean stability during storage differ both between temperatures and form of preparation (lysate or purified). Phage lysates at 4 °C remained active up to day 147th. At room temperature, the preparation was stable until day 68, a slight decrease in the titer was observed on day 27, while on day 147 no active phages were observed in the lysate. In the purified preparation, a decrease in the titer of active phage particles was observed at each of the tested temperatures. On the 27th and 68th day of the experiment, the decrease in phage titer, both for room temperature and at 4 °C, remained at the same level (~two orders of magnitude). On day 147, the preparation kept at room temperature completely lost its activity, while the mixture of purified phages stored at 4 °C did not change its activity. The preparation at 35 °C already showed no active phage particles by the 27th day of the experiment (Figure 3). One of the research aims was to select a safe and attractive carrier for phage preparations for administration to bees. A 50% sucrose solution was tested along with preparations P1 (1/A), P2 (3/A) and P3 (1/A + 3/A), which were administered to bees in laboratory experiments. The stability of the P3 preparation at two temperatures: 22 °C (room temperature) and 4 °C was used to verify the optimal storage conditions of preparation intended for potential future use in apiaries. At the 4 °C, the addition of sucrose allows phage activity to remain for 8 months of the experiment (there was a decrease in the mean titer by about two orders of magnitude), while at room temperature, after 3 months of the experiment, the mean titer of the studied phages dropped (Figure 4). Additionally, the mean titer of all phage preparations suspended in royal jelly decreased sharply in the first day of the experiment (decreased by three orders of magnitude). On the remaining days, the mean titer remained constant (Figure 5).

The optimization of amplification effectiveness of the *Paenibacillus*-specific bacteriophage indicated that, in most cases, the optimal phage to bacterial ratio was 1:100 or 1:1000. For the studied phages, the best results were obtained at 34 °C/0 RPM/18 h. The conducted research shows that for the tested phage the effectiveness of its amplification depends on the strain on which it is amplified, knowing the growth dynamics of the host strain, and therefore the phage culture conditions were selected individually. The obtained results show that the phage 2/A had the fastest adsorption rate, while the 1/A phage had the lowest when host strain no. 408 was used (Table 5).

Phage preparations without the cryoprotectant lost their activity on the 40th day of the experiment (a decrease by two to three orders of magnitude was observed). On day 125, mean phage titer was 10 PFU/mL for 2/A, 98.25 × 10^3^ PFU/mL for 1/A, 4 × 10^2^ PFU/mL for 3/A, 1.2 × 10^3^ PFU/mL for 5/A and 4 × 10^2^ PFU/mL for 1/A + 3/A (decrease by four to five orders of magnitude) (Figure 6C). Among the preparations with the addition of sugars, the most stable were those with the addition of sucrose (compared to the positive control preparations with glucose) which is presented in the Appendix A. The preparation containing phage 1/A and 3/A was particularly advantageous because in both cases the mean phage titer was of the same order of magnitude as before lyophilization (Figure 6A,B). The mean titers of the preparations with the addition of 10% sucrose decreased in each of the samples by about an order of magnitude (Figure 6A).

### 3.1. Phage DNA Isolation, Sequencing and Bioinformatics Analyses of Sequenced Data, Protein Profile of Studied Phages

The DNA isolated from the tested phages and bacteria was high-quality, free from contaminating RNA, proteins or solvents (Appendix A). De novo analysis was performed for 1/A, 2/A, 3/A, 4/A and 5/A phages. Tested phages belonging to *Caudovirales* and their DNA is double-stranded. The content of G + C pairs for individual phages is as follows: 1/A—41.5%, 2/A—41.5%, 3/A—41.6%, 4/A—41.6%, 5/A—41.5% (according to: https://www.sciencebuddies.org/science-fair-projects/references/genomics-g-c-content-calculator, 15 January 2019). The sizes of the genomes of studied phages are presented in Table 6.

Each phage was compared to the homologous virus in the NCBI database (GenBank) and to the Diva phage (NC_028788.1) which is a typical representative of the *Sitaravirus*. The phages were compared with each other, showing no significant differences (98.01% similarity). Phages 1/A, 4/A and 5/A are genetically similar to *Paenibacillus* phage Xenia (NC_028837.1), while phages 2/A and 3/A to *Paenibacillus* phage Shelly (KP296795.1). Shelly and Xenia phage genomes were used as reference genomes in de novo assembly of sequences (Figure 7).

The protein profiles obtained for phages 1/A, 2/A, 3/A and 5/A were very similar to each other, which confirm the data obtained by bioinformatic analysis (Appendix A). From the performed analysis, 64 amino acid readings were obtained that were repeated for all tested bacteriophages. 25 peptides were qualified as unique sequences for the estimated phages. These results were compared with the NCBI database. Eight amino acid sequences form obtained phages were identified in Xenia and Shelly (Table 7). This result confirms that the studied phages: 1/A, 4/A and 5/A are genetically similar to already known *Paenibacillus* phage Xenia (NC_028837.1) [61], and phages 2/A and 3/A to *Paenibacillus* phage Shelly (KP296795.1) [62], with 98% similarity.

For all tested phages, the presence of integrases in the genome was confirmed, which may indicate a temperate character [63]. The integrases were located towards the end of the genome and preceded the region of the genes influencing the host’s virulence. In the case of the 5/A bacteriophage, there is a potential point mutation in the integrase that calls into question the correct functioning of this gene. In the course of the analyses, in each of the phages, a region located in the end parts of the genome was detected, in which there are four genes that may significantly affect the virulence of the host or indicate the temperate nature of the bacteriophage. The region includes: phospho-manno-mutase, the toxin-antitoxin system (hicA, hicB) and HNH endonuclease. Comparative analysis of the sequences obtained as a result of annotation with the shared fragments of *Paenibacillus* 408 and 453 host sequences did not show any significant homology between the studied bacteriophages and their hosts. Full annotation tables (Appendix A) with genomic maps of studied phages (Appendix A) are presented in the Appendix A.

### 3.2. Phage In Vivo Study

During the experiment there were no negative effects of phage cocktail on the mean feed intake in the group of bees fed sugar syrup with an addition of phage cocktails. In the group fed sugar syrup with addition of phage cocktails, mean feed intake was 38.99 ± 12.67 and 35.6 ± 3.85 µL/bee/day for P3-1% and P3-10%, respectively. These amounts were not significantly different in comparison to other groups. The lowest mean feed intake was recorded in the control group at 32.39 ± 5.71 µL/bee/day. In turn, the highest syrup consumption was noted in the P2-10% group (46.85 ± 11.63 µL/bee/day) and P1-10% group (43.86 ± 5.59 µL/bee/day) and the differences compared to the control group were statistically significant (Figure 8).

There were significant differences in bee mortality between the group fed a diet supplemented with 10% of phage 3/A and the control group. The highest and the lowest value of this parameter was 25.32 ± 11.97% and 7.57 ± 2.06% for P2-10% and K, respectively. Among the groups receiving the diet containing 1% or 10% of phage supplements the lowest mortality was noted in P3-1% and P3-10% group was 8.91 ± 7.77% and 16.03 ± 5.60%, respectively. Feed intake showed no significant differences between these groups and the control (Figure 9). The most common behavior within groups was walking: on average 15.21 times per group. In contrast, the least frequent behavior was immobility, which on average occurred 1.77 times per group. There were no significant differences between the control group and experimental groups in mean time and total duration of walking (Figure 10 and Figure 11).

Based on results from studies on bees in laboratory conditions, a phage cocktail at the highest concentration was selected for in vivo assays which were performed to assess the potential negative effect of the phage cocktails on bee larvae. The cocktail caused a slight decrease in survival compared to controls, but the decrease was not statistically significant. On the seventh day of the treatment, the mean survival of the larvae fed with diet supplemented with the phage cocktail in a concentration of 10% (LP3) did not differ from the control group and was 70%, compared with K1 and K2 which were on average 78.75% and 76.25%, respectively.

## 4. Discussion

The observed phenomenon of growing bacterial resistance has led to the necessity of developing new therapeutic strategies, e.g., with the use of bacteriophages, which have become an alternative to antibiotic therapy in the world crisis of antibiotic effectiveness [64]. The problem occurs both in humans and in animals. This may be avoided when phages especially in cocktail form containing phages of different lytic spectra, is applied [65]. Therefore, we developed a phage preparation which contains two phages specific to *P. larvae*. The selection of phages for the final preparation was based on the phage lytic spectrum, their ability to amplify as well as their stability during storage (after addition of a stabilizer).

For phage isolation biological and environmental samples (e.g., soil, water, wax, bees, honey) obtained from apiaries and their surroundings were used. Because of the slow growth of these strains and the possibility of coexistence of other strains, competing pathogenic bacterial strains in the obtained bee wax, it was likely that we could obtain other strains with faster and more expansive growth [66], such as *Escherichia coli*, which could lead to isolation of the latter. At the same time, no AFB pathogens would be detected, and the result of isolating these strains would be a false negative. Moreover, the hosts of the isolated phages are environmental strains with specific non-standard culture requirements, so it was necessary to optimize the phage amplification process to each phage individually to obtain the highest phage titer in the preparation needed for further studies. These factors probably contributed to isolation of only five phages from the whole tested material over 2500 samples.

Phage application in apiaries in vivo should be preceded by their detailed characterization (phage activity, lytic spectrum, life cycle parameters, genome sequences, phage stability under expected conditions at the site of application or infection) and testing of their effectiveness in vitro as stated by Jończyk-Matysiak et al. (2020) [23]. Our phages have been characterized and described (as shown in Figure 12) before we decided to choose phages for the final preparation composition. The lytic spectrum of our phages was 31–46%, whereas, e.g., Ribeiro et al. (2019) described the API480 phage isolated from a hive soil sample which proved to be active against 69% of the tested field of *P. larvae* strains in vitro [33]. The conducted research has shown that the studied phages show specificity only for the strains causing AFB (*Paenibacillus* spp.), while not destroying bacteria constituting the natural intestinal microbiota of bees (bacteria isolated from the intestines of healthy bees in which no phages were used showed no sensitivity to the phages we tested) which suggests that these phages can be safely used.

Examination of phage adsorption indicated that even after 10 min of contact with bacteria over 99% of phage particles were adsorbed to its host (for phage 1/A, 2/A, 5/A) whereas in similar studies over 85% adsorption was observed after several minutes [33].

Bacteriophage 4/A was not stable both at room temperature and under refrigeration; moreover, there were problems with amplification and obtaining the appropriate titer for this phage. This phage was characterized with a broader lytic spectrum than others. Due to the existing problems, bacteriophage 4/A was excluded from the remaining stability studies and was not taken into account in the selection of components of the final preparation. Since phages 1/A, 4/A and 5/A and phages 2/A and 3/A turned out to be genetically similar, it was also finally decided that a cocktail containing phages 1/A and 3/A. Phages 1/A and 3/A were selected to compose a phage cocktail (which would be prepared for cage and field studies) because they belonged to different genetic types of *Paenibacillus* phages. What is more, the similarity analysis indicates that these phages are not new, but have been already characterized [61,62]. The results of our research indicated that isolated phages are temperate. This is in accordance with published data demonstrating the most phages against *P. larvae* are indeed temperate (encoding integrases or transposases) [22,26,67], but current state of the knowledge does not exclude this type of phage in phage therapy in the fight against AFB [39]. However, Ribeiro et al. (2019) have not identified integrase gene and lysogeny module in genome of the mentioned API480 phage [33]. During the analyses, in each of the studied phages, a region was detected on the end of the genome in which there are four genes that may significantly affect the host’s virulence or suggest the temperate nature of the studied bacteriophages. The region includes: phospho-manno-mutase (PMM), toxin–antitoxin system (hicA, hicB) and HNH endonuclease. PMM is involved in the biosynthesis of bacterial exopolysaccharides that form biofilms, protecting cells against the action of antibiotics and environmental factors [68]. TA-type systems are also found in studied phages. After integrating into the host genome, the bacteriophage with this virulence factor increases the virulence of the bacteria. HicA and hicB code for the complete bioprotein TA system, influencing the mRNA translation process in the bacterial cell. The HicA protein acts as an interferase that cleaves mRNA molecules, significantly affecting the rate of translation in the cell. HicB neutralizes HicA by acting as an antitoxin [69]. Endonucleases are widely distributed among phages (especially *Caudovirales*) and are usually responsible for point catalyzing genomic DNA hydrolysis. Following hydrolysis, HNH endonuclease are very often introduced as introns at the DNA cleavage site [70]. The specific activity of HNH proteins can significantly protect DNA against the action of restriction enzymes, which may prove its value as a lysogen [71,72]. No literature reports link antibiotic resistance with other genes identified in the phages tested.

Interestingly, the presence of phages active against *Paenibacillus* was detected in samples of the intestines of bees which were given the phages at titers ranging from 10^2^ to 10^4^ PFU/mL (data not presented herein). This proves that the phages may penetrate into the intestinal lumen after consumption by bees, i.e., to the site of infection, where *Paenibacillus* strains amplify, and may retain their activity after oral administration (data not shown). Likewise, Ribeiro et al. (2019) [73] have shown in an in vivo study the ability of an active phage to penetrate larvae after per os administration of adult honeybees. They applied a T7 phage suspended in 50% (*w*/*v*) sucrose, similar to our experiments. In their phage biodistribution assay, they also proved phage penetration through the larval midgut epithelium, which confirmed that phages could be active at the site of *P. larvae* infection. This is credible evidence supporting the safety of the tested phages when administered orally to honeybees.

As was proven, different factors may influence phage activity and efficacy: physico-chemical conditions, storage, host individual conditions, preparation composition, titer, route of phage administration, and phage inability to penetrate and achieve high concentration at the site of expected action [74]. An important issue for phage antibacterial application is the form of phage formulation and its storage stability. The obtained results prove that the most favorable storage temperature for the studied phage preparations is 4 °C; moreover, the crude preparations retain their activity longer, which is related to the favorable “environment” consisting of bacterial residues and the culture medium. According to our results, the purification process of phage lysates which significantly decreased the level of bacterial endotoxins in preparations was significant compared to their level in the concentrated lysate, which indicates that this process was highly effective, and might reduce the risk of phage intolerance by bees when using the preparation. Moreover, the elimination of bacterial contaminants could also contribute to the fact that bees were more likely to take such a preparation, but our results indicate that this type of preparation is characterized with worse stability during storage. As is shown in this paper, phage lysates are the most optimal environment for bacteriophages, as they contain bacterial residues and appropriate concentrations of salt and other nutrients. Under these conditions bacteriophages will be most stable. Stability tests of phage preparations in the form of dialyzed PBS lysates were carried out on preparations that were applied in bee experiments. An effective solution to prevent a drop in phage titer in a preparation is to add the preparation at an appropriate concentration just before administering the preparation to bees. The half-life of the preparation will be longer; moreover, when mixed with sucrose, it will be readily consumed by bees in a shorter time. The 10% P3 preparation suspended in 50% sugar syrup proving to be the best solution for phage stability as well as tolerability by bees. Moreover, as bee products should contain as much sugar as possible, it seemed reasonable to use 50% solutions. From among all sugars examined, it was decided that sucrose, as the most common sugar and the cheapest, will be the most widely used, and is often used by beekeepers to feed bees.

Low phage concentrations and difficulties with their availability [63] may not be sufficient to reduce the count of *P.*
*larvae* to prevent and cure AFB and may constitute other imitations of phage action. Therefore, preparations with high phage titers, which contain substance protecting phages against harmful hive conditions (temperature, humidity, pH, larval food composition) and known stability under hive conditions, should be applied. Our preparation applied to bees had a final titer in the range of 10^3^–10^5^ PFU/mL, but the short period of exposition to harmful hive conditions could not cause phage prompt inactivation. The checked stability of phages in the P3 cocktail in sucrose under varied temperature and acidic pH in hive proved the possible potential of phages to be active when they penetrate to the site of infection-hive midgut. According to data obtained in experiments in which tested phages were incubated in different pH values (data not presented herein), it was observed that they retained their activity at pH = 5 after 21 days of incubation (the observed drop in their titer was by one–two orders of magnitude) which may seem sufficient for use in beehive conditions. It was shown that phages in formulation with royal jelly intended to feed larvae were inactivated by the latter because of its low pH, as well as its composition [30,33]. Furthermore, honey completely inactivated phages in vitro [75,76]. These bee products could be one of the possible reasons for failure of phage application in bees suffering from AFB. These findings are consistent with our observations in which royal jelly is a product that causes rapid phage inactivation and this product is not appropriate to apply phages.

Bacteriophages are viruses composed of proteins and nucleic acids, therefore the lyophilization process seems to be a good method of stabilizing them for long-term storage. Lyophilization of proteins can provide more stability through reducing the molecular mobility, hydrolysis and contamination, but physical and chemical instabilities are not clear. During protein lyophilization, it is important to consider their conformation (protein structure), and chemical and physical stability as well as their ability to completely dissolve upon reconstitution [52]. The lyophilization process is usually performed with the addition of cryoprotectant compounds and our research indicates that the addition of sucrose to the phage formulation was preferred, and in particular, the formulation 1/A + 3/A with the addition of sucrose.

In the present study the safety of different phage preparations was tested in laboratory experiments. The results obtained in the cage experiment on adult bees showed that the two different doses of phage cocktail (1% and 10%) suspended in 50% sugar syrup added and incorporated into the diet of adult bees did not influence bee mean mortality and mean feed intake, which allows the inference that it did not have a negative influence on bee health. This also may confirm that there is no negative effect on the bee microbiota ecosystem. The results obtained in experimental groups were similar compared with the control group. Comparison of behaviors between the control group and treated groups allow exclusion of negative effects of the phage cocktail on honeybee workers. Checking the influence of phages on adult bees is justified due to the exclusion of possible toxic effects of the additive on the insect’s organism and the effectiveness of distributing the preparation over the colony. This is justified because any preparation used in the treatment or prophylaxis of bees, above all, must not harm them. It is also important that the preparation reaches every bee in the colony, as the prophylaxis must include all individuals. Due to these features, the research was mainly narrowed down to various types of sugars, as only sweet products are attractive and eagerly eaten by bees (as sweet as possible, because bees can taste flavors and the sweet taste is best tolerated), and choosing the additives that on the one hand would protect phage activity but on the other contain as few ingredients in the final preparation as possible.

Larvae survival was slightly lower in the group receiving a diet with a phage cocktail addition compared to the control, but this difference was not significant. These results are consistent with results of other authors [27,30]. Compared to the present study, Ghorbani-Nezami et al. (2015) [39] observed a slightly higher value of survival rates for a negative control (84.4%) and the F, WA, and XIII phage control larvae (88.8%, 85.5%, and 86.6%, respectively), but they did not demonstrate a statistically significant difference among groups. The results also indicate that administering the phages to larvae does not cause any adverse effect to larval survival and can be considered a safe addition to their diet. It can be assumed that a slight decrease observed in the percent of larvae survival fed with phage cocktails suspended in PBS may not be directly due to the action of phages but may result from nutrient dilution. The obtained results showed that phage preparation is safe and well tolerated by bees, which may indicate a protective or therapeutic effect.

In the course of the present studies one additional phage with a lytic spectrum broader than the described phages was isolated (data not presented herein) whose safety and therapeutic potential against AFB will be evaluated further. The results presented in this report, strongly suggest phage safety and tolerability when applied to bees. Further studies on the use of phage therapy in the fight against AFB are warranted. In our opinion the studies both presented herein and planned are needed because of the importance of the role of honey bees both in the ecosystem and in food production globally [77,78,79].

## Figures and Tables

**Figure 1 viruses-13-01217-f001:**
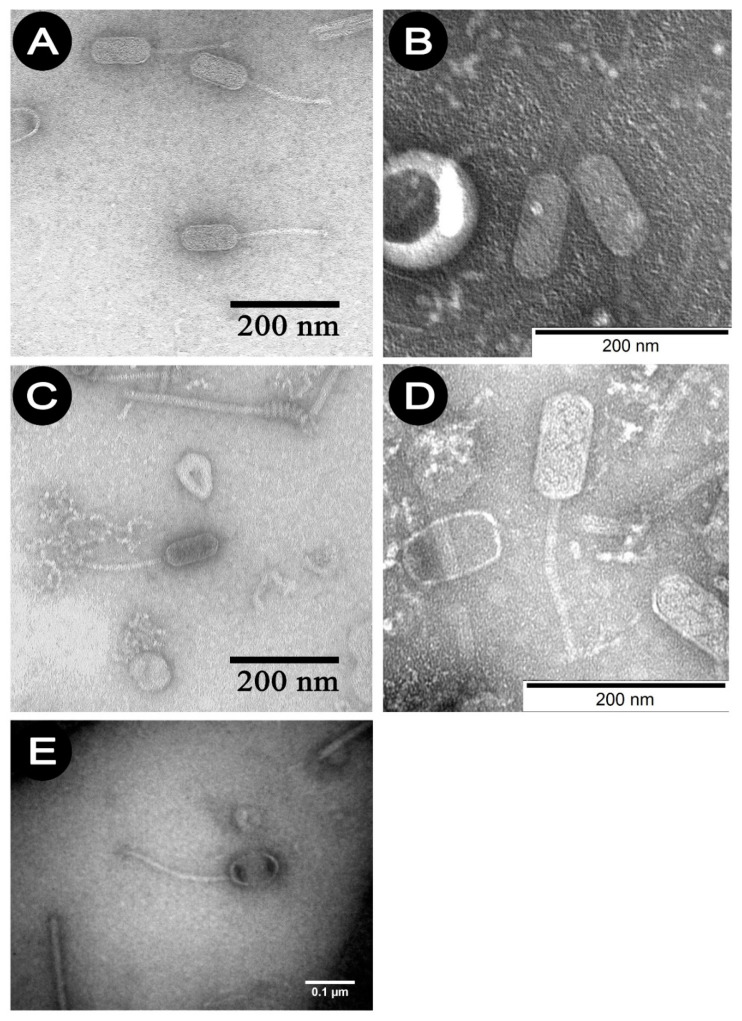
Micrographs in transmission electron microscopy of phage. (**A**) 1/A, (**B**) 2/A, (**C**) 3/A, (**D**) 4/A, (**E**) 5/A.

**Figure 2 viruses-13-01217-f002:**
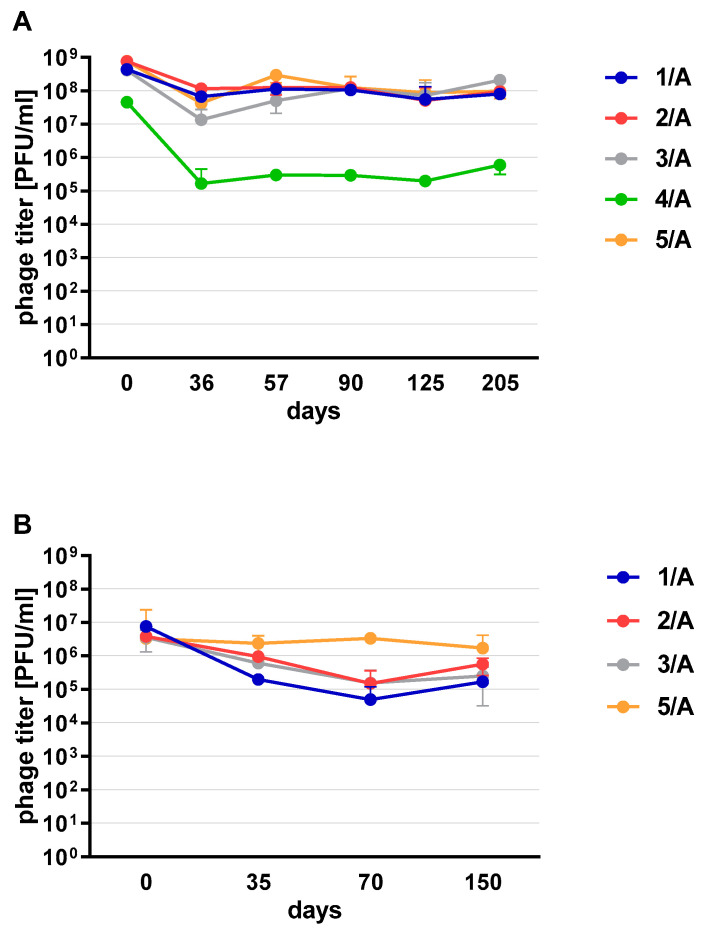
Phage stability at 4 °C in (**A**) lysate (**B**) purified preparation. Error bars represent standard deviation (±SD) of mean phage titer.

**Figure 3 viruses-13-01217-f003:**
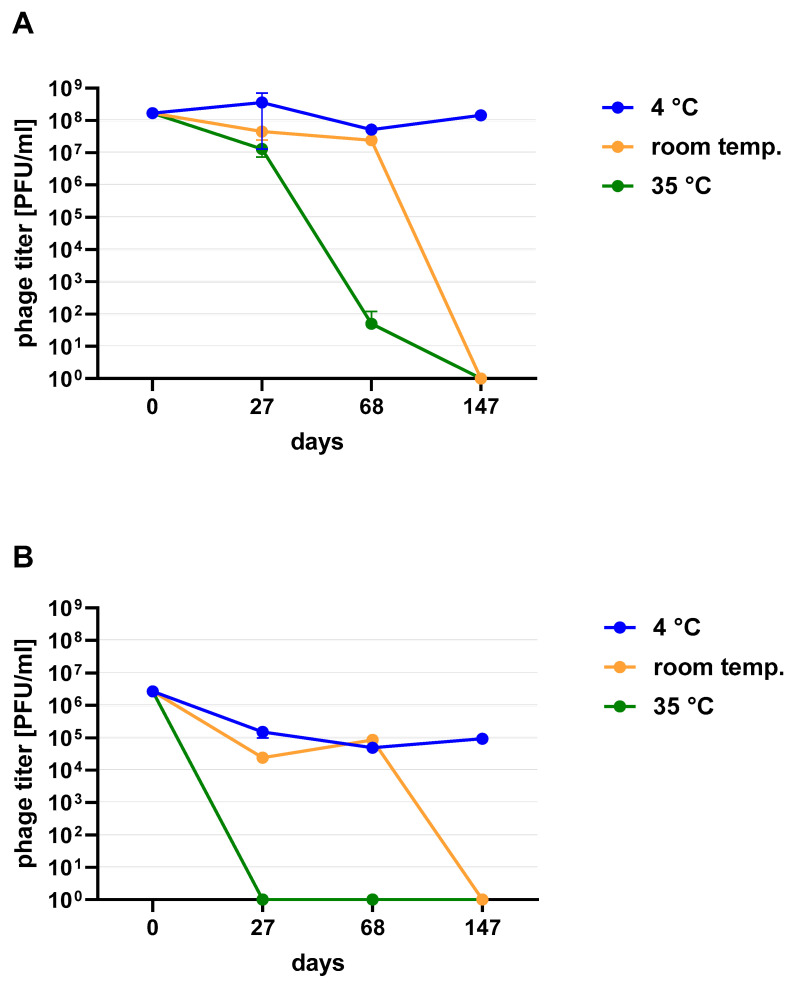
Stability of phage cocktail in (**A**) lysate (**B**) purified preparation stored at 4 °C, room temperature and 35 °C. Error bars represent standard deviation (±SD) of mean phage titer.

**Figure 4 viruses-13-01217-f004:**
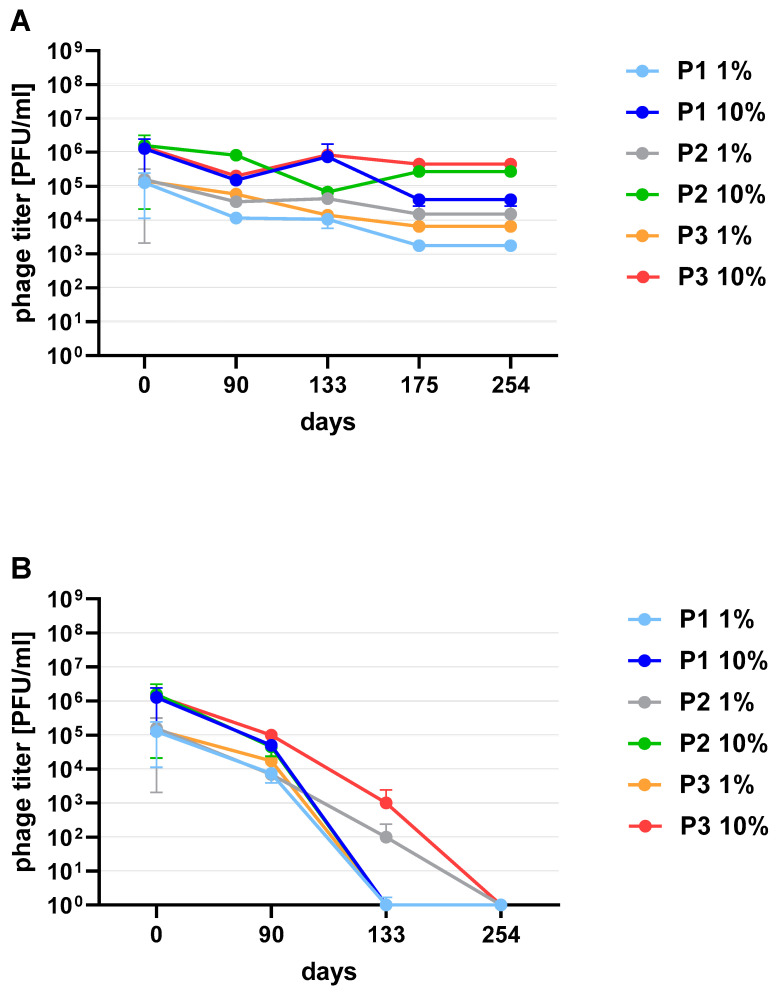
The stability of 1% and 10% phage preparations: P1 (1/A), P2 (3/A) and P3 (1/A + 3/A) suspended in 50% sucrose (**A**) at 4 °C; (**B**) room temperature. Error bars represent standard deviation (±SD) of mean phage titer.

**Figure 5 viruses-13-01217-f005:**
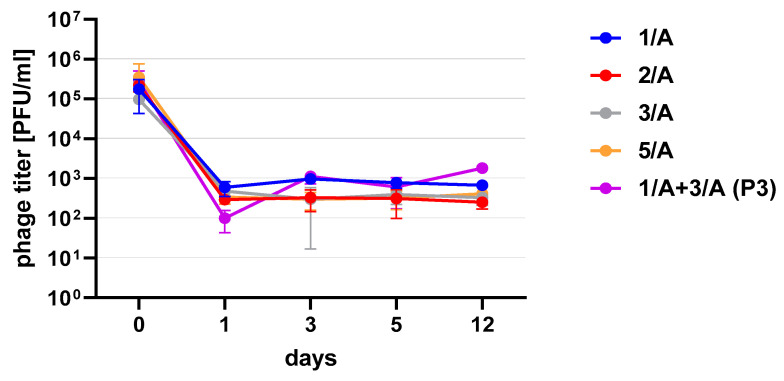
The mean stability (±SD) of phage preparations in 5% royal jelly. Error bars represent standard deviation (±SD) of mean phage titer.

**Figure 6 viruses-13-01217-f006:**
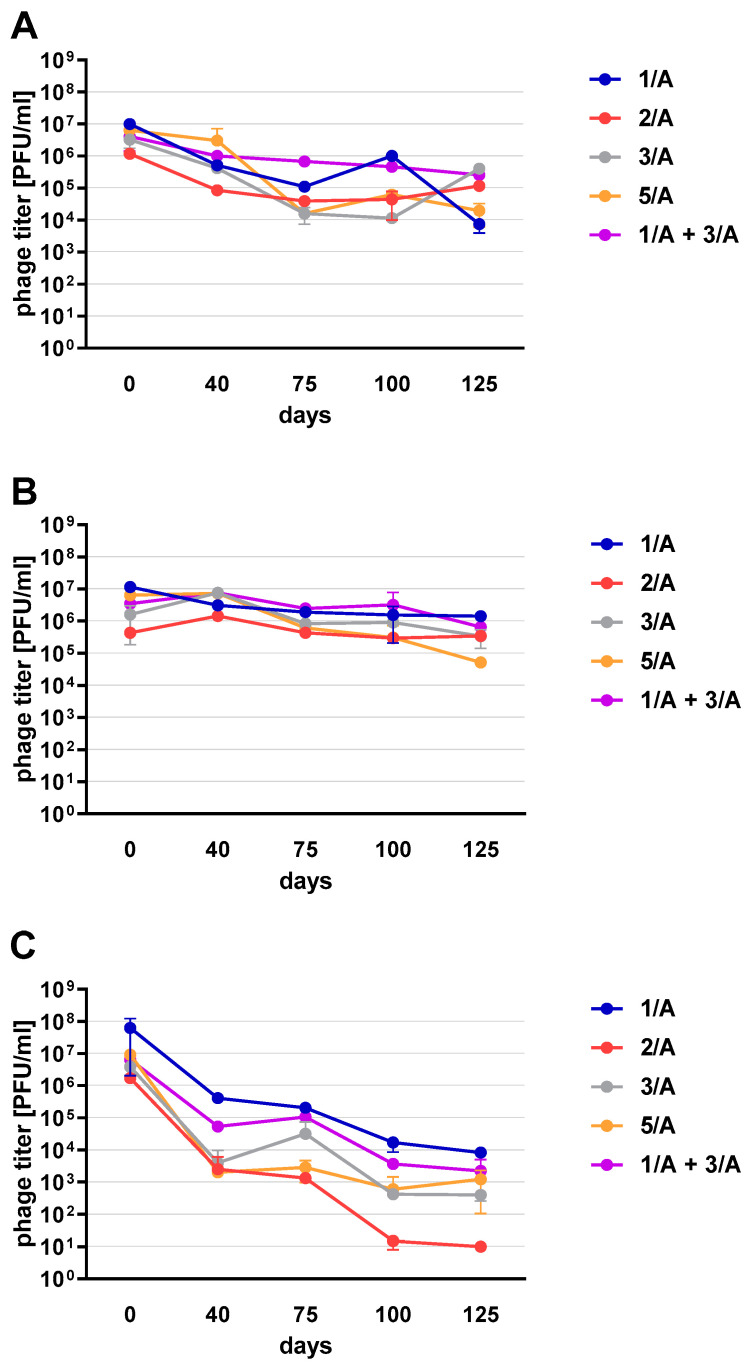
The stability of lyophilized phages with (**A**) 10% sucrose (**B**) 10% trehalose (**C**) without cryoprotectant. Error bars represent standard deviation (±SD) of mean phage titer.

**Figure 7 viruses-13-01217-f007:**
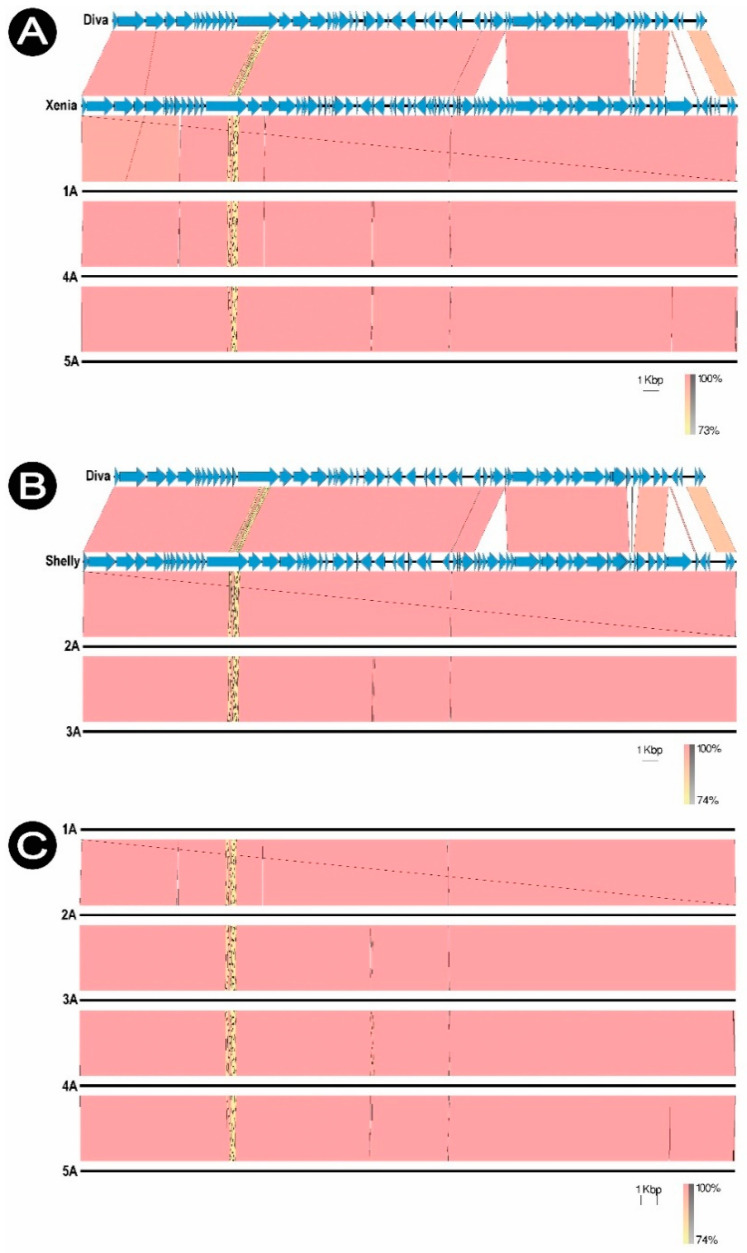
Comparison of the genomic sequences. (**A**) 1/A, 4/A and 5/A, (**B**) 2/A and 3/A, (**C**) 1/A, 2/A, 3/A, 4/A, 5/A *Paenibacillus* phages.

**Figure 8 viruses-13-01217-f008:**
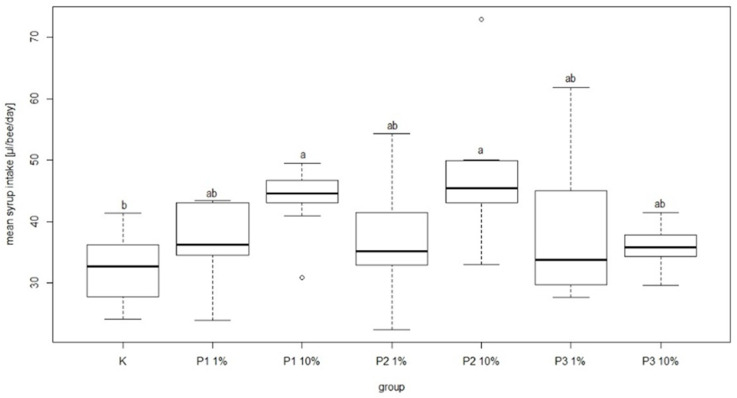
Mean syrup intake (µL/bee/day). Each box represents a different group. Black horizontal lines and vertical bars represent medians and quartiles for nine cages. Bars with different superscript letters represent a significant difference *p* < 0.05, a, b—significance of differences on the level *p* < 0.05. K—control group fed only with 50% sugar syrup; group P1-1% —fed sugar syrup with an addition of 1% of phage 1/A, group P1-10%—fed sugar syrup with an addition of 10% of phage 1/A, group P2-1%—fed sugar syrup with supplementation of 1% of phage 3/A, P2-10%—fed sugar syrup with supplementation of 10% of phage 2/A, P3-1%—fed with sugar syrup with an addition of 1% of phage cocktail 1/A + 3/A, P3-10%—fed with sugar syrup with an addition of 1% of phage cocktail 1/A + 3/A.

**Figure 9 viruses-13-01217-f009:**
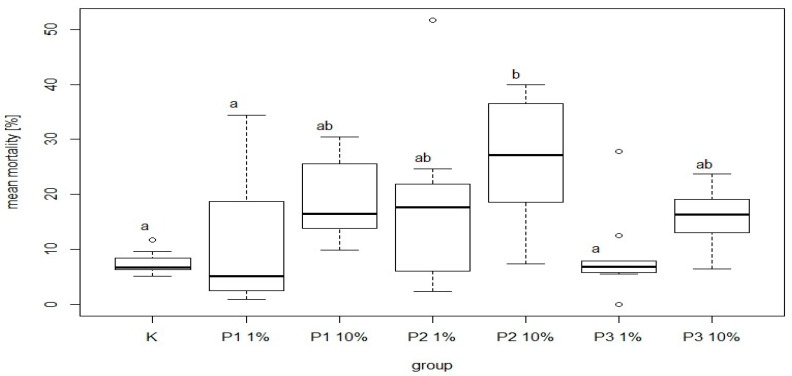
Mean bee mortality (%). Each box represents a different group. Black horizontal lines and vertical bars represent medians and quartiles for nine cages. Bars with different superscript letters represent a significant difference *p* < 0.05, a, b—significance of differences on the level *p* < 0.05. K—control group fed only with 50% sugar syrup; group P1-1%—fed sugar syrup with an addition of 1% of phage 1/A, group P1-10%—fed sugar syrup with an addition of 10% of phage 1/A, group P2-1%—fed sugar syrup with supplementation of 1% of phage 3/A, P2-10%—fed sugar syrup with supplementation of 10% of phage 2/A, P3-1%—fed with sugar syrup with an addition of 1% of phage cocktail 1/A + 3/A, P3-10%—fed with sugar syrup with an addition of 1% of phage cocktail 1/A + 3/A.

**Figure 10 viruses-13-01217-f010:**
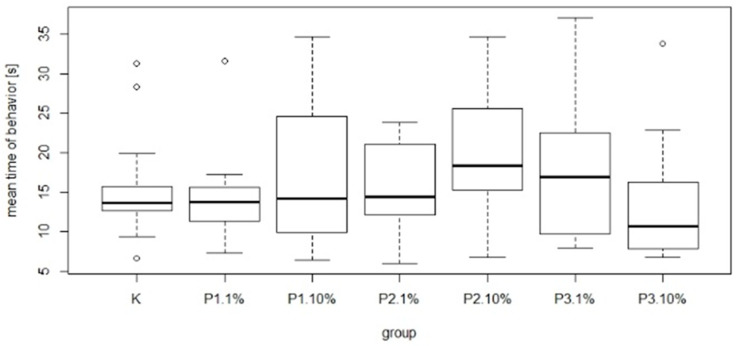
Mean time of walking (s). Each box represents a different group. Black horizontal lines and vertical bars represent medians and quartiles for seven cages. K—control group fed only with 50% sugar syrup; group P1-1%—fed sugar syrup with an addition of 1% of phage 1/A, group P1-10%—fed sugar syrup with an addition of 10% of phage 1/A, group P2-1%—fed sugar syrup with supplementation of 1% of phage 3/A, P2-10%—fed sugar syrup with supplementation of 10% of phage 2/A, P3-1%—fed with sugar syrup with an addition of 1% of phage cocktail 1/A + 3/A, P3-10%—fed with sugar syrup with an addition of 1% of phage cocktail 1/A + 3/A.

**Figure 11 viruses-13-01217-f011:**
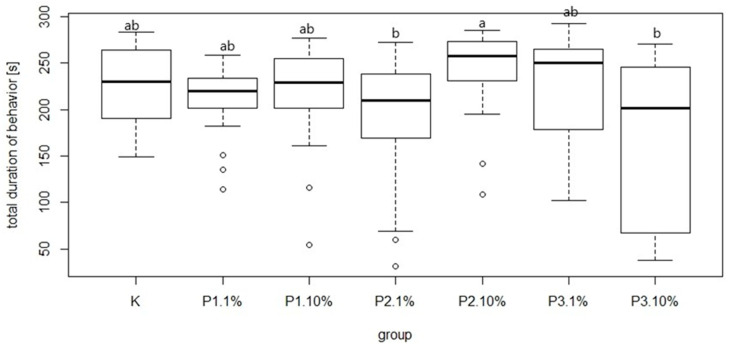
Total duration of walking (s). Each box represents a different group. Black horizontal lines and vertical bars represent medians and quartiles for seven cages. Bars with different superscript letters represent a significant difference *p* < 0.05, a, b—significance of differences on the level *p* < 0.05. K—control group fed only with 50% sugar syrup; group P1-1%—fed sugar syrup with an addition of 1% of phage 1/A, group P1-10%—fed sugar syrup with an addition of 10% of phage 1/A, group P2-1%—fed sugar syrup with supplementation of 1% of phage 3/A, P2-10%—fed sugar syrup with supplementation of 10% of phage 2/A, P3-1%—fed with sugar syrup with an addition of 1% of phage cocktail 1/A + 3/A, P3-10%—fed with sugar syrup with an addition of 1% of phage cocktail 1/A + 3/A.

**Figure 12 viruses-13-01217-f012:**
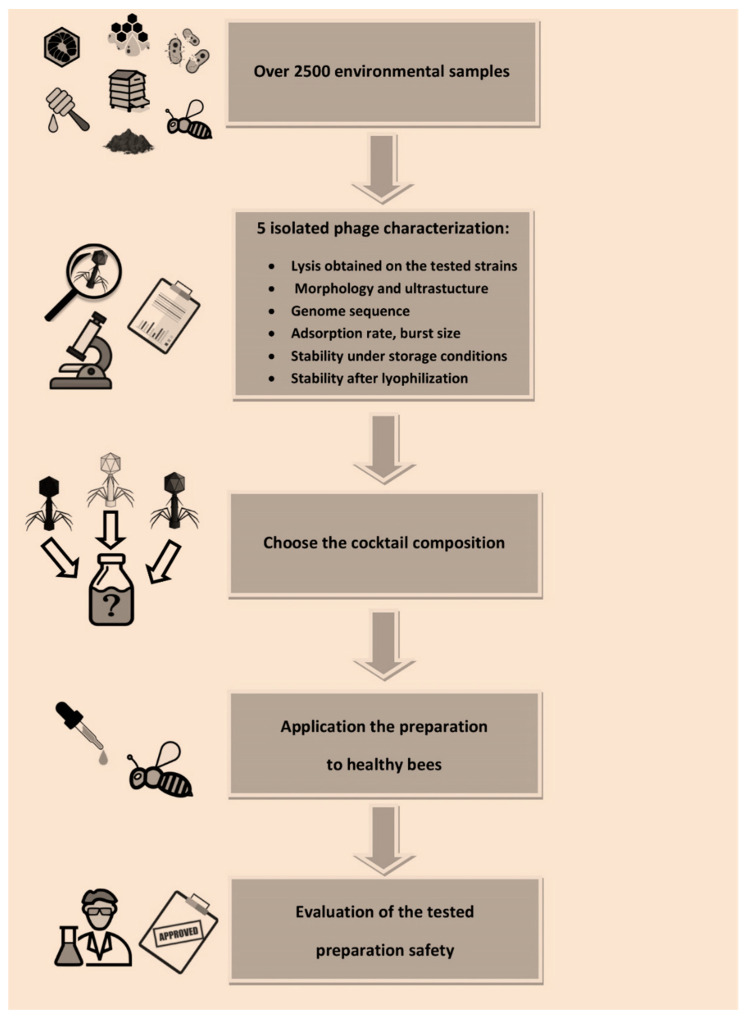
Scheme of the development of the phage cocktail intended for honeybee treatment.

**Table 1 viruses-13-01217-t001:** List of isolated strains of *Paenibacillus* spp.

Strain Number	Name of Bacterial Strain	Source of Isolation
1	*Paenibacillus larvae* 412 *	bee brood
2	*Paenibacillus larvae* 451	bee brood
3	*Paenibacillus larvae* 405	bee brood
4	*Paenibacillus larvae* 448	bee brood
5	*Paenibacillus larvae* 416	bee brood
6	*Paenibacillus larvae* 453	bee brood
7	*Paenibacillus larvae* 695	wax comb
8	*Paenibacillus larvae* 1328	wax comb
9	*Paenibacillus larvae* 1346	wax comb
10	*Paenibacillus larvae* 1562	wax comb
11	*Paenibacillus larvae* 1390	wax comb
12	*Paenibacillus larvae* 1434	wax comb
13	*Paenibacillus larvae* 1576	wax comb
14	*Paenibacillus larvae* 1598	wax comb
15	*Paenibacillus larvae* 1432	wax comb
16	*Paenibacillus larvae* 1772	wax comb
17	*Paenibacillus larvae* ATCC 846	reference strain
18	*Paenibacillus pulvifaciens* ATCC 49843	reference strain
19	*Paenibacillus larvae* 1695C	wax comb
20	*Paenibacillus larvae* 1602	wax comb
21	*Paenibacillus pabuli* 1591	wax comb
22	*Paenibacillus pabuli* 272	wax comb
23	*Paenibacillus thiaminolyticus* 408	bee brood
24	*Paenibacillus larvae* 440	bee brood
25	*Paenibacillus alvei* 418	bee brood
26	*Paenibacillus larvae* 1537	bees
27	*Paenibacillus larvae* 2149B	wax comb
28	*Paenibacillus chinjnensis* 2162	wax comb
29	*Paenibacillus larvae* 2152	wax comb
30	*Paenibacillus larvae* 2168	wax comb
31	*Paenibacillus larvae* 2154	wax comb
32	*Paenibacillus larvae* 2150	wax comb
33	*Paenibacillus larvae* 2149	wax comb
34	*Paenibacillus larvae* 2155	wax comb
35	*Paenibacillus larvae* 1856	bees

* The number next to the species name corresponds to the number of the sample from which the strain was isolated.

**Table 2 viruses-13-01217-t002:** Sources of phage isolation.

Phage ICTV Symbol	Phage Symbol	Source of Phage Isolation	The Location the Sample Came from
*Paenibacillus* phage vB_PlaS-1/A	1/A	wax comb	Świętokrzyskie Province
*Paenibacillus* phage vB_PlaS-2/A	2/A	wax comb	Lower Silesia
*Paenibacillus* phage vB_PlaS-3/A	3/A	bees	Lower Silesia
*Paenibacillus* phage vB_PlaS-4/A	4/A	wax comb	Świętokrzyskie Province
*Paenibacillus* phage vB_PlaS-5/A	5/A	wax comb	Lubuskie Province

**Table 3 viruses-13-01217-t003:** Morphological characteristics of *P. larvae* phages.

Phage Symbol	Family	Morphotype	Dimensions [nm]	
			Head	Tail
1/A	*Siphoviridae*	B3	120 × 50	170
2/A	*Siphoviridae*	B3	115 × 40	135
3/A	*Siphoviridae*	B3	150 × 45	190
4/A	*Siphoviridae*	B3	115 × 55	170
5/A	*Siphoviridae*	B2	100 × 70	220

**Table 4 viruses-13-01217-t004:** Level of bacterial endotoxins in phage lysate before and after purification.

No.	Phage Symbol	Endotoxin Concentration in Phage Lysate (EU/mL)	Endotoxin Concentration after Preparation Purification (EU/mL)
1	1/A	2.646	0.450
2	2/A	2.570	0.873
3	3/A	1.810	0.246
4	4/A	3.610	2.500
5	5/A	2.500	0.350

**Table 5 viruses-13-01217-t005:** Phage adsorption rate to host bacteria.

Phage Symbol	Host Bacteria	Mean Adsorption Rate (±SD) after 10 min (%)	Mean Adsorption Rate (±SD) after 20 min (%)	Mean Adsorption Rate (±SD) after 40 min (%)
**1/A**	408	19.44 ± 12.53	27.22 ± 6.15	42.78 ± 11.68
**1/A**	453	97.98 ± 1.10	99.02 ± 0.53	99.31 ± 0.20
**2/A**	408	1.77 ± 7.29	68.57 ± 7.71	86.43 ± 3.75
**2/A**	453	99.91 ± 0.08	99.95 ± 0.01	99.99 ± 0.01
**3/A**	408	72.50 ± 6.29	77.50 ± 6.17	78.00 ± 5.34
**3/A**	453	85.00 ± 5.51	90.01 ± 3.99	94.15 ± 1.19
**4/A**	408	59.25 ± 8.28	81.05 ± 0.44	81.35 ± 1.11
**4/A**	453	63.25 ± 7.18	64.00 ± 3.77	83.92 ± 2.13
**5/A**	408	97.57 ± 0.78	98.12 ± 0.72	98.38 ± 0.52
**5/A**	453	92.02 ± 0.04	95.48 ± 3.47	97.16 ± 5.00

**Table 6 viruses-13-01217-t006:** Size of studied bacteriophage genomes and number of detected genes.

Phage Symbol	Genome Size (bp)	The Number of Detected Genes
1/A	41 052	69
2/A	41 177	68
3/A	41 209	67
4/A	41 147	67
5/A	41 111	67

**Table 7 viruses-13-01217-t007:** Identified peptide sequences from phages 1/A, 2/A, 3/A and 5/A and their matching to the NCBI database.

Sequence	Protein from *Paenibacillus* Phage Xenia (NC_028837.1)	Protein from *Paenibacillus* Phage Shelly (KP296795.1)	Protein Length (aa)	Protein Mass (Da)
TGEEHMSFYK	Major capsid protein YP_009201915.1″	Major capsid protein AJK27925.1	376 aa	Xenia (41,710.35) Shelly (41,740.38)
EMKADGDTVTFGR				
FLPKTVSTDILVEPTVK				
MKTLYEL				
TVSTDILVEPTVKNPLR				
KEMKADGDTVTFGRN				
KGGSGDFSTWFGRRF	Portal protein YP_009201913.1	Portal protein AJK27923.1	Xenia (411 aa) Shelly (417 aa)	Xenia (47,675.28) Shelly (47,111.62)
KCEYPGCKKTAQETFALVPLCKW	Hypothetical protein YP_009201975.1″	Hypothetical protein AJK27978.1	71 aa	8392.90

## Data Availability

The presented data are available on request from the corresponding author.

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
