# Peer review of "Isolation and Characterization of Phages Active against Paenibacillus larvae Causing American Foulbrood in Honeybees in Poland"

_viruses, 2021, doi:10.3390/v13071217_

Round 1
Reviewer 1 Report
The manuscript of Jonczyk-Matysiak and co-workers presents isolation and characterization of phages active against Paenibacillus larvae, causing American Foulbrood in honey bees in Poland. AFB is a devastating disease of the beehives, causing the death of cultivated honey bees worldwide.
The topic is relevant for bee farmers as well as it is interesting scientifically. The authors hypothesize that the use of bacteriophages against the AFB causing agent (P. larvae) may be an innovative and scientifically sound measure to combat the disease. However, this idea is not new; many papers have been already published using phages against P. larvae (for example, doi: 10.1186/1471-2164-15-745., doi: 10.1016/j.jip.2017.09.010. – this field is saturated, and the presented manuscript does not add any new information to the topic. Likewise, the methods used by the authors are all standard – the study presents a typical course of experiments concerning phage isolation and characterization (including phylogeny, host specificity, efficacy tests etc.).
Due to the nature of the topic and description and characterization of phages against P. larvae in literature, I judge this manuscript of low interests to readers.
Also, I found it really strange that this sort of paper is authored by 23 authors (sic!) – it looks like everyone does everything, but the overall workload seems not to be that big. This long list of authors gives me some ethical concerns about preparing the manuscript, which the authors should give feedback on.
All graphs lack standard error or standard deviations bars – they should be included in the graphs. Also, it is not clear how many times the experiments were repeated, which is essential to judge the quality of the data. Figures 7 A and 7 B possess a low quality.
Author Response
We thank for valuable comments, and we would like to apologize for some inaccuracies in the text of the manuscript. According to our opinion the topic presented in our manuscript is interesting and needed because of role of honey bees both in ecosystem and due to the importance of bees in food production all over the world, and the following papers confirm the value of bees: Patel et al., 2020; Porto et al., 2020; Lippert et al., 2021 (newly added reference: 77-79).We agree that the methods described in our manuscript are standard, however, we do not consider this as a negative factor. The potential of phages in combating AFB has been emphasized in line: 89-92, and the importance to conduct further studies which would confirm their safety, activity as well as effectivity in field conditions. What is more, these studies can enable the commercialization of a preparation that would be widely available to beekeepers whose applicability of therapeutic agents to combat AFB is currently very limited and still remains which an unsolved problem. A review by Tsourkas et al, 2020 (reference 22) also strongly emphasizes that the topic of using phages in the context of foulbrood disease in wide context of research is still needed and actually.
We agree with the opinion of the reviewer, the number of co-authors seems too numerous. The studies that have been presented in our manuscript came from the project that was conducted in cooperation between four centers: two research centers and two companies, whose employees made an intellectual contribution during preparation of this paper. In order to confirm this fact, we highlight the contribution of each of the co-authors in more detail below:
EJM - the author of the research idea and article concept, drafted the main part of the manuscript, verification and analysis of obtained results, preparation revised version of the manuscript;
BO- contributed to the part of the manuscript, phage amplification experiments conducting, preparation puiricafation, evaluation the level of endotoxins in tested preparations, preparation phage cocktails for using in bees experiments and phage genome sequencing;
EP - contributed to the parts of the manuscript, designed and prepared experiments on bees, keeping animals for laboratory tests, support during preparation revised version of the manuscript;
KŚJ- contributed to the part of the manuscript, preparation phage genome sequencing, bioinformatics analyses of sequence data;
PM – collecting material for studies, prepared experiments on bees, keeping animals for laboratory tests;
MC- contributed to the parts of the manuscript, support during preparation revised version of the manuscript, preparing Figures and reference list;
NŁ- contributed to the part of the manuscript, figures preparation, optimization phage amplification method, prepared material for TEM analysis
DK - contributed to the part of the manuscript, prepared isolation of bacterial strains and searching phages in procured material, tested the stability of phages
JN - contributed to the part of the manuscript, isolation of bacterial strains and searching phages in procured material, studied phage lytic spectrum, phage specificity, isolation bacteria from bee gut, searched the presence of the applied P. larvae phages in gut of bees treated with phages;
KHS - contributed to the part of the manuscript, prepared experiments on phage stability, searching appropriate career to final preparation composition;
MK- contributed to the part of the manuscript, prepared MALDI-TOF experiments;
FO - contributed to the part of the manuscript, prepared isolation of bacterial strains and searching phages in procured material, prophage induction from bacterial strains experiments, optimization phage amplification;
NB - contributed to the part of the manuscript, isolation of bacterial strains and searching phages in tested samples prepared adsorption experiments;
AM - prepared Figures, data analysis;
AB- contributed to the part of the manuscript, study the presence of phage cocktail residues in bee products;
MS - contributed to the parts of the manuscript, study the presence of phage cocktail in bee products;
BB - contributed to the parts of the manuscript isolation of bacterial strains and searching phages in tested material,
WF- contributed to the parts of the manuscript procurement material for bacterial strains and phage isolation, support during preparation revised version of the manuscript;
SL – contributed to the parts of the manuscript procurement material for phage isolation from bees;
PC - contributed to the parts of the manuscript, keeping animals for laboratory tests, collecting samples, designed experiments non bees;
BWD- contributed to the parts of the manuscript give conceptual advices regarding phage isolation method and phage amplification to obtain higher titer in the preparations,
AR- designed experiments non bees and gave conceptual advices at bees experiments, sample collection
AG contributed to the parts of the manuscript, provided support and conceptual advice at all stages of manuscript preparation.
All authors had full access to all the data in the study and take responsibility for the integrity of the data and the accuracy of the data analysis. We hope that current explanation of authors contribution will now be sufficient for the reviewer.
Unfortunately, we do not have gels pictures of a higher quality, therefore we decided to move this Figure to supplementary materials.
We hope that our current corrections are acceptable and make the improvement of the manuscript`s value.
Reviewer 2 Report
First of all, some editorial work will be needed due to letter size, graphs, and tables. Figure 13 is colouring is completely misleading.
Figure 7 should be in supplementary material, other figures should be aligned and combined. many information from figure and tables are repeated in main text.
Also, a discussion is a big result repetition.
Author Response
We thank for valuable comments, and we would like to apologize for some inaccuracies in the text of the manuscript. All suggested editorial corrections have been made using the “Track Changes” function. Size of letters in the main text was unified. Colours of the Figure 12 have been changed to be more subdued. Two tables and one figure were deleted, because their content was similar to information in the main text of the manuscript. Discussion has been rephrased to be more coherent. We hope that our current corrections are acceptable and make the improvement of the manuscript`s value.
Round 2
Reviewer 1 Report
The revised version of the manuscript has been significantly improved compared to the previous version, and the Authors have answered all raised comments. I think that now the manuscript should be accepted as it is for publication in the journal.
Reviewer 2 Report
the authors addressed all comments